# Improvement of Skin Condition Through RXR Alpha-Activating Materials

**DOI:** 10.3390/biom15020296

**Published:** 2025-02-17

**Authors:** Sanghyun Ye, Seonju Lee, Seongsu Kang, Seung-Hyun Jun, Nae-Gyu Kang

**Affiliations:** LG Household and Health Care R&D Center, Seoul 07795, Republic of Korea; shye123@lghnh.com (S.Y.); seonju@lghnh.com (S.L.); franck.kang@lghnh.com (S.K.)

**Keywords:** skin aging, RXR, wrinkle improvement, anti-inflammation, anti-oxidant

## Abstract

Retinol is well-known anti-aging material in the cosmetics industry, owing to its proven superior efficacy both in vitro and in vivo. Despite its high efficacy, retinol is associated with limitations, such as skin irritation and its potential photodegradation. Retinol is converted into retinoid acid within cells, which then exerts a cellular response by activating both the retinoic acid receptor (RAR) and retinoid x receptor (RXR). Noting that RAR activity is associated with skin irritation and RXR activation alone can enhance skin-related indicators without inducing inflammation, we developed an alternative approach for skin anti-aging focusing solely on RXR activation. We found that combined treatment of andrographolide and *Bidens pilosa* extract successfully activated RXR alpha and enhanced *RXRA* gene expression. Moreover, we investigated their efficacy using dermal fibroblasts and keratinocytes and found that they enhanced the gene expression of extracellular matrix (ECM) proteins with anti-oxidant and anti-inflammation efficacies. Finally, in a human clinical trial, we confirmed that our materials successfully improved wrinkles in various areas, skin elasticity and hydration without causing irritating side effects. These findings highlight the potential of our RXR alpha-activating materials as an anti-wrinkle solution that avoids the typical side effects associated with retinol.

## 1. Introduction

Skin aging is a complex phenomenon caused by various intrinsic and extrinsic factors [1]. Extrinsic aging is primarily caused by environmental factors, such as prolonged exposure to ultraviolet (UV) radiation, air pollution, and dietary habits [2]. Notably, UV exposure, especially from sunlight, accelerates the breakdown of collagen and elastin, leading to wrinkles, sagging skin, and age spots [2]. In contrast, intrinsic aging is a genetically programmed process that reflects an individual’s genetic makeup [3]. Factors, such as genetics, hormonal changes, and natural degenerative processes, also play important roles in intrinsic aging [1,3].

With the increasing demand for youthful and healthy skin, the cosmetics industry has developed numerous skincare products. Among these, retinol stands out as a widely recognized and powerful anti-aging ingredient [4,5]. Extensive research has highlighted its benefits, such as reducing facial wrinkles and enhancing skin elasticity through topical application [5,6,7]. However, the use of retinol comes with limitations, including the potential to cause skin irritation and its potential photodegradation [8,9]. To address these issues, researchers have focused on strategies like using lower retinol concentrations or combining it with soothing or hydrating ingredients [10,11]. These approaches aim to preserve retinol’s efficacy while minimizing irritation. Despite these efforts, the above-mentioned limitations still persist, prompting the ongoing search for powerful anti-aging cosmetics with reduced irritation.

Retinol undergoes metabolic transformations within cells and is ultimately converted into retinaldehyde and retinoic acid via various pathways [12,13]. Retinoic acid, which exists in different isoforms, such as all-trans retinoic acid (ATRA) and 9-cis retinoic acid, plays a significant role in providing benefits of retinol on the skin [13,14]. ATRA, the most physiologically active form of retinoic acid in the skin, activates the retinoic acid receptor (RAR) [14]. In contrast, 9-cis retinoic acid serves as a high-affinity ligand for the retinoid X receptor (RXR) [15]. RXR, an essential nuclear receptor, functions as a transcription factor and regulates cell cycle, metabolism, and cell death by partnering with various receptors [16]. RXR has been reported to be expressed five times more in human skin than RAR [17]. Nevertheless, most research on compounds, such as ATRA and retinol, has focused more on the role of RAR, despite RXR being highly associated with biological responses, which remains to be clearly understood [7,18].

Treatment with ATRA and RAR agonists can lead to upregulated collagen type 1 (COL1A1) and collagen type 3 (COL3A1) expression while mitigating UV-induced damage to collagen, resulting in enhanced collagen content in photoaged skin [19,20]. However, ATRA treatment often causes skin irritation mediated by RAR activity [21,22]. In contrast, treatment with 9-cis retinoic acid or RXR agonist increases epidermal thickness, suggesting that RXR signaling may be involved in skin barrier function and homeostasis [23,24]. Additionally, RXR agonists have been found to inhibit inflammatory condition, with low levels of RXR activity closely associated with heightened inflammatory response and atopic dermatitis [25,26,27]. Despite these findings, RXR agonists have predominantly been investigated in cancer treatment, with limited research on their potential in the cosmetic industry [28,29]. These insights suggest that targeting RXR may be a viable approach for developing effective skincare products that deliver significant benefits with minimal irritation.

In recent years, extensive research has been conducted to discover natural ingredients with powerful anti-aging effects similar to those of retinol [30,31]. Some studies have focused on screening compounds with functional or structural similarities to retinol [32]. One such compound is andrographolide, derived from *Andrographis paniculata*, which shares a diterpenoid structure with retinol [33,34]. It can exhibit anti-aging properties as well as other benefits, such as anti-oxidant, anti-inflammatory, and anti-cancer effects [35,36]. The anti-cancer activity of andrographolide operates independent of RAR activity, potentially through RXR activity [36].

Another herb, *Bidens pilosa*, is widely known for its anti-inflammatory and anti-oxidant effects [37,38,39]. *B. pilosa* extract (BPE) contains phytol, a diterpenoid compound that activates RXR after being converted to phytanic acid within cells [39,40,41]. Given these properties, we investigated the potential of andrographolide and BPE.

In this study, we explored whether the combined treatment of andrographolide and BPE can activate RXR alpha and assessed the dependence of their efficacy on skin cell lines on RXR alpha activity. To investigate their mechanism of action and efficacies, we conducted experiments using skin cell lines, dermal fibroblasts, and keratinocytes and confirmed their skin-improving effects through a human clinical trial.

## 2. Materials and Methods

### 2.1. Cell Culture and Preparation

Human skin keratinocytes (HaCaT) and human skin fibroblasts (Hs68) were purchased from AddexBio (T0020001, San Diego, CA, USA) and the American Type Culture Collection (CRL-1635, Manassas, VA, USA), respectively. Hs68 cells were cultured using DMEM (Gibco, Grand Island, NY, USA) supplemented with 10% FBS (Gibco) and penicillin–streptomycin (Gibco) at 37 °C with 5% CO_2_. HaCaT cells were cultured in a customized calcium-free DMEM (Solbio, Seoul, Republic of Korea) supplemented with 10% FBS, penicillin–streptomycin (Gibco), 1 mM sodium pyruvate (Gibco), 2 mM L-glutamine (Gibco), and 0.01 mM CaCl_2_ (Sigma-Aldrich, St. Louis, MO, USA) at 37 °C with 5% CO_2_. At least three independent experiments on cell lines were executed.

Inoformation about materials and their working concentrations was listed in the Table 1. Andrographolide and BPE were procured from Alphacryptec (Cheongwon, Republic of Korea) and Chemyunion (São Paulo, Brazil), respectively. Retinol, ATRA, 9-cis retinoic acid, and the RXR antagonist (UVI3003) were purchased from Sigma-Aldrich. The RAR antagonist (AGN193109) was obtained from R&D Systems (Minneapolis, MN, USA). All substances were diluted in DMSO to make a stock solution, then diluted in cell culture media and treated with cells. The working concentrations of the materials used in this experiment were determined through separate experiments.

**Table 1 biomolecules-15-00296-t001:** List of materials and their concentrations.

Material	Concentration
Andrographolide	1 μg/mL
*Bidens pilosa* extract	10 μg/mL
Retinol	10 μM
All-trans retinoic acid	1 μM
9-cis retinoic acid	1 μM
UVI3003	10 μM
AGN193109	50 μM

### 2.2. RAR-γ and RXR-α Activation Assays

To evaluate RAR-γ activity, RAR-γ reporter cell lines were acquired from BPS Bioscience (San Diego, CA, USA). RAR-γ reporter cells were cultured in DMEM (Gibco) supplemented with 10% FBS (Gibco), 1% penicillin–streptomycin (Gibco), 400 µg/mL geneticin (G418) (Invitrogen, Waltham, MA, USA), 1 µg/mL puromycin (Hyclone, Logan, UT, USA), and 100 μg/mL hygromycin (Hyclone) at 37 °C with 5% CO_2_. The cells were seeded in black 96-well plates (33396, SPL life sciences, Pocheon, Republic of Korea) and incubated overnight before being treated with test compounds for 24 h. The luciferase activity of RAR-γ was analyzed by quantifying it using the Steady-Glo luciferase assay system (Promega, Madison, WI, USA). Luciferase activity was measured by luminescence fiber optics using a Synergy H1 microplate reader (BioTek Instruments Inc., Winooski, VT, USA).

The RXR alpha activation assay was performed using the GeneBLAzer™ RXR alpha HEK 293T DA assay kit (Invitrogen), according to the manufacturer’s instructions. Briefly, cells were seeded in black 96-well plates (SPL sciences) and incubated for 24 h. After overnight incubation, the cells were treated with test compounds for 24 h. Then, the LiveBLAzer™-FRET B/G (CCF4-AM) substrate included in the kit was added to each well. After 2 h of incubation at room temperature, fluorescence intensity was measured using a Synergy H1 microplate reader (BioTek Instruments Inc.). Fluorescence in the blue channel was measured with an excitation filter (409/20 nm) and emission filter (460/40 nm). The FRET signal in the green channel was measured with the excitation filter (409/20 nm) and emission filter (530/30 nm). RXR alpha activity was quantified by dividing the blue emission values by the green emission values (460:530 nm).

### 2.3. Quantitative Real-Time PCR (RT-qPCR)

HaCaT and Hs68 were seeded into 6-well plates at a density of 3 × 10^5^ cells per well and incubated for 24 h at 37 °C. The test compounds were then added in a serum-free medium for another 24 h. The AccuPrep^®^ Universal RNA Extraction Kit (Bioneer, Daejeon, Republic of Korea) was used to extract total RNA. The concentration and purity of the extracted RNA were determined by the absorbance at wavelength 230, 260 and 280 nm using a Nanodrop spectrometer (Thermo Fisher Scientific, Waltham, MA, USA). cDNA was synthesized using a cDNA synthesis kit (Philekorea, Seoul, Republic of Korea) and 1 μg of total RNA. The synthesis was performed according to the manufacturer’s instructions and carried out using the Veriti 96 Well Thermal Cycler (Applied Biosystems, Foster City, CA, USA). The following conditions were applied for reverse transcription: 42 °C for 30 min and 72 °C for 10 min. RT-qPCR was performed with TaqMan™ Universal PCR Master Mix (Applied Biosystems) and Power SYBR Green PCR Master Mix (Applied Biosystems) using the StepOnePlusTM RT-PCR system (Applied Biosystems), according to the manufacturer’s protocol. RT-qPCR was performed using the following commercial TaqMan primers (Thermo Fisher) and designed primers listed in the Table 2: *GAPDH* (Hs02786624_g1), *COL1A1* (Hs00164004_m1), *COL4A1* (Hs00266237_m1), *COL5A1* (Hs00609133_m1), *COL7A1* (Hs00164310_m1), *ELN* (Hs00899658_m1), *FBN1* (Hs00171191_m1), and *FN1* (Hs01549976_m1).

**Table 2 biomolecules-15-00296-t002:** List of primer sequences designed for target mRNA.

Target mRNA	Forward Primer	Reverse Primer
*GAPDH*	CATGTTCGTCATGGGGTGAACCA	AGTGATGGCATGGACTGTGGTCAT
*RXRA*	TGCTTCGTGTAAGCAAGTACATAAG	CTCTTTATGGATCTGTCATCCTCTC
*RARA*	CAGAGCAGCAGTTCTGAAGAGATA	GACACGTGTACACCATGTTCTTCT
*IL-6*	AGACAGCCACTCACCTCTTCAG	TTCTGCCAGTGCCTCTTTGCTG

### 2.4. Enzyme-Linked Immunosorbent Assay (ELISA)

For assessing pro-collagen I alpha 1, total MMP1, and IL-8, Hs68 were seeded into 24-well plates at a density of 2 × 10^4^ cells per well and cultured for 24 h. Then, the test compounds were added to serum-free medium and incubated with the cells for another 24 h. Culture supernatant was collected to measure the secreted target protein using the Human Pro-Collagen I alpha 1 DuoSet ELISA kit (R&D Systems), Human Total MMP-1 DuoSet ELISA kit (R&D Systems), and Human IL-8/CXCL8 DuoSet ELISA (R&D Systems). All experiments were performed according to the manufacturer’s instructions.

For IL-1 alpha, HaCaT cells were seeded into 24-well plates at a density of 2 × 10^4^ cells per well and cultured for 24 h. The cells were irradiated once with UV-B at an intensity of 30 mJ/cm^2^ using BIO-SUN irradiation system (Vilber Lourmat, Marne-la-Vallée, France) and treated with test compounds for an additional 24 h. Culture supernatant was collected to measure secreted IL-1 alpha using the Human IL-1 alpha/IL-1F1 DuoSet ELISA kit (R&D Systems).

Protein levels were adjusted to total protein concentration, which was quantified using the Pierce™ BCA Protein Assay Kit (Thermo Fisher).

### 2.5. Reconstructed Three-Dimensional (3D) Human Skin

The reconstructed 3D human skin model Neoderm^®^-ED was purchased from Tego Science (Seoul, Republic of Korea) and cultured following the manufacturer’s instructions. The control cream formulation consists of following ingredients: distilled water, tromethamine, carbomer (2-propenoic acid, polymer with 2,2-bis(hydroxymethyl)propane-1,3-diol 2-propenyl ether), xanthan gum, EDTA-3Na, 1,2-hexanediol, betaine, glycerin, dipropylene glycol, isocetyl myristate, dimethicone/vinyl dimethicone crosspolymer, cyclopentasiloxane, cyclohexasiloxane, squalene, caprylic/capric triglyceride, lecithin, C12–20 alkyl glucoside, C14–22 alcohols, beeswax, ceteareth-20, PEG-40 stearate, cetearyl alcohol, glyceryl stearate, stearyl alcohol and cetyl stearyl alcohol. The cream used in this study was formulated according to the protocols set by the LG H&H (Seoul, Republic of Korea). Test cream formulations containing andrographolide and BPE were topically applied, with a control cream lacking these extracts. After 2 d of incubation at 37 °C with 5% CO_2_, the 3D skin was fixed with 4% paraformaldehyde and stained with Masson’s trichrome. Section images were captured using the EVOS FL Auto2 Imaging System (Thermo Fisher Scientific). Dermal collagen content was analyzed using ImageJ Software version 1.54 (NIH, Bethesda, MD, USA).

### 2.6. Immunocytochemistry for Fibrillin-1

Hs68 cells were seeded into 24-well plates at a density of 1 × 10^4^ cells per well and cultured for 24 h. The culture media were changed with PBS, and cells were irradiated once with UV-B at an intensity of 30 mJ/cm^2^ using the BIO-SUN irradiation system (Vilber Lourmat). The test compound were added to serum-free medium and treated to the cells for another 24 h. Cells were fixed with 4% paraformaldehyde and permeabilized using 0.1% Triton X-100 (Sigma-Aldrich) in PBS. After washing three times with PBS, the cells were blocked with 5% FBS in PBS for 30 min at room temperature and incubated with diluted anti-fibrillin 1 antibody (ab53076, 1:250; Abcam, Cambridge, UK) at 4 °C overnight. After washing three times with PBS, the cells were incubated with a secondary antibody Goat Anti-Rabbit IgG H&L (ab150077, 1:1000, Abcam) for 1 h at room temperature. For nuclear staining, the samples were stained with diluted 4′,6-diamidino-2-phenylindole (DAPI) solution (5 mg/mL, Sigma-Aldrich) for 10 min. After washing thrice with PBS, fluorescent images were captured using the EVOS™ FL Auto2 Imaging System (Thermo Fisher Scientific) and analyzed using Image J Software version 1.54 (NIH).

### 2.7. Measurement of Cellular Viability and Reactive Oxygen Species (ROS)

For cellular viability test, Hs68 cells were seeded in black flat- and clear-bottom 96-well plates at a density of 1 × 10^4^ cells per well and cultured for 24 h. The cells were then irradiated once with UVB at an intensity of 30 mJ/cm^2^ using the BIO-SUN irradiation system (Vilber Lourmat) or treated with 800 μM H_2_O_2_ to induce cell death. After 24 h of incubation with the test compounds, cell viability was determined using the Cell Counting Kit-8 cell proliferation assay (Dojindo Molecular Technologies Inc., Kumamoto, Japan), according to the manufacturer’s instructions.

For cellular oxidative stress analysis, the 2′,7′-dichlorofluorescein diacetate (DCFDA) assay was performed using the DCFDA/H2DCFDA—Cellular ROS Assay Kit (Abcam) according to manufacturer’s instructions. Briefly, the cells were seeded in black flat- and clear-bottom 96-well plates at a density of 1 × 10^4^ cells/well and incubated for 24 h. After incubation, the cells were treated with test materials for another 24 h. Cellular ROS levels were stained by adding 20 μM DCFDA solution for 3 h. The cells were then treated with 800 μM H_2_O_2_ solution for 30 min. Fluorescence was measured using a Synergy H1 microplate reader (BioTek Instruments Inc.) at Ex/Em = 485/535 nm.

### 2.8. Human Clinical Trial

This study was approved by the Ethics Committee of the LG H&H Institutional Review Board (LGHH-20230720-AA-03-01). Fourteen healthy Korean volunteers (aged 26–50 years; mean age, 34.43 years) were recruited for this clinical trial. Pregnant women, those receiving skin treatments at clinics, minors and those who did not have the ability to speak Korean were excluded. All participants were informed of the possible side effects, and written informed consent was obtained before the trial began. The tests were performed in a half-face, double-blind manner. The base cream was made as mentioned in Section 2.5. The test cream formulation contained 0.035% andrographolide and 0.35% BPE, whereas the control cream contained 0.1% retinol. All participants were instructed to use both creams for 4 weeks.

To assess skin irritation, a self-evaluation guideline was employed from a prior study [11]. Briefly, participants applied the test cream formulation on the left side of their face and control cream on their other side for three consecutive days. Five categories (desquamation, pruritus, burning, dryness, and stinging) were self-evaluated daily on a scale of 0 to 3, depending on the degree of irritation. The cumulative scores over the 3 d were used to determine the degree of skin irritation.

Prior to measurements, all participants were instructed to wash their faces and rest for a minimum of 20 min in a humidity (45 ± 5%)- and temperature (22 ± 2 °C)-controlled room. Facial wrinkles were measured in triplicate using Antera 3D camera (Miravex, Dublin, Ireland). Changes in fine wrinkles under the eyes, and crow’s feet were analyzed using the roughness (Ra) texture parameter, while nasolabial folds were analyzed based on the fold length parameter in the Antera 3D program (Miravex). Skin elasticity was assessed by measuring R2 and R7 parameters using Cutometer^®^ MPA 580 (C + K Electronic GmbH, Cologne, Germany). Skin hydration was measured using the capacitance-based Corneometer CM 825 (C + K Electronic GmbH). All measurements were performed according to the manufacturer’s instructions. Participants who did not have any initial wrinkles in specific areas at week 0 were excluded from the analysis.

### 2.9. Statistical Analysis

Data are presented as mean ± standard error of the mean (SEM) values derived from at least three independent experiments. Before conducting the analysis for statistical significance, we first checked if the data followed a normal distribution. If the data followed a normal distribution, we used a *t*-test or one-way ANOVA analysis to determine statistical significance. If the data did not follow a normal distribution, we used a nonparametric test for our analysis. Differences were considered statistically significant at * *p* < 0.05, ** *p* < 0.01, and *** *p* < 0.001. The data were analyzed using GraphPad Prism version 6.07 (GraphPad Software, La Jolla, CA, USA).

## 3. Results and Discussions

### 3.1. RXR Alpha Activation by Combined Treatment of Andrographolide and BPE

We investigated whether andrographolide and BPE could successfully activate RXR alpha without activating RAR. Our findings revealed a slight increase in RXR alpha activity when andrographolide or BPE was used alone (Figure 1a). However, when both were applied simultaneously, RXR alpha activity increased more significantly than in their individual use (Figure 1a). Furthermore, we confirmed that andrographolide and BPE did not activate RAR gamma and showed no synergistic effects when they were combined (Figure 1b). Positive controls were used for each assay to verify their accuracy. 9-cis retinoic acid successfully activated RXR alpha, whereas retinol and ATRA successfully activated RAR gamma (Figure 1a,b).

To investigate whether andrographolide and BPE affected the expression of receptor genes, RT-qPCR was conducted. We observed that andrographolide and BPE synergistically increased *RXRA* gene expression (Figure 1c). To assess the impact of andrographolide and BPE on subunits of RAR other than RAR gamma, we verified any changes in the *RARA* gene expression and confirmed that these materials did not influence the expression of the *RARA* gene (Figure 1d). RXR agonists were previously reported to not affect *RXRA* gene expression [24]. Interestingly, in our study, andrographolide and BPE not only enhanced RXR alpha activity but also upregulated the expression of *RXRA*. These findings suggest that the combined action of andrographolide and BPE can synergistically influence RXR alpha activity.

Considering the results of both receptor activity and gene expression assays, the combined treatment of andrographolide and BPE sufficiently activated RXR alpha without affecting RAR gamma activity.

### 3.2. Collagen Synthesis Effects of Combined Treatment of Andrographolide and BPE in the Skin Cells

Skin aging is accompanied by a decrease in collagen content and an increase in MMP1 expression in the dermis [42]. Therefore, we used human dermal fibroblasts (Hs68) to evaluate the collagen enhancing effects of andrographolide and BPE. Our results indicated that treatment with andrographolide alone increased procollagen I alpha 1 to a degree similar to that observed with retinol treatment (Figure 2a). However, when andrographolide and BPE were combined, we observed an even greater increase in procollagen I alpha 1 level than that in retinol treatment alone (Figure 2a). Furthermore, simultaneous treatment with both compounds led to a significant decrease in MMP1 levels (Figure 2b). The RXR agonist bexarotene can reduce MMP1 levels, likely through the activation of the p38 mitogen-activated protein kinase/nuclear factor-κB pathway [26]. Thus, further studies are required to determine whether our complex activates the same pathway to decrease MMP1 levels.

To examine whether the co-treatment of andrographolide and BPE influences other types of collagen, Hs68 was treated with the materials for 24 h and gene expression was analyzed using RT-qPCR. The combined treatment significantly increased *COL1A1*, *COL4A1*, *COL5A1,* and *COL7A1* gene expression (Figure 2c). There was no significant difference between the combined treatment group and retinol treatment group (Figure 2c).

Considering the complexity of human skin, which is composed of diverse cell types and has a sophisticated structure, we also assessed the efficacy of andrographolide and BPE using reconstructed 3D human skin. We applied a cream containing andrographolide and BPE to 3D human skin for 2 d and performed Masson’s trichrome staining. We found that the co-treatment of andrographolide and BPE significantly enhanced collagen content in the dermis of 3D skin compared to the control cream (Figure 2d,e). Taken together, our results suggest that the combined treatment of andrographolide and BPE is effective in strengthening the dermis by increasing collagen content.

### 3.3. Efficacies of Combined Treatment of Andrographolide and BPE on Extracellular Matrix (ECM) Component Enhancement

In addition to collagen, the dermis consists of various ECM proteins [43]. For example, elastic fibers, majorly composed of elastin and fibrillin-1, play crucial role in maintaining the integrity and skin elasticity of the skin [44,45]. Fibronectin is another key ECM protein that interacts with other ECM components to maintain structural stability [46]. A recent study highlighted the role of RXR signaling in ECM turnover across various organs [47]. Therefore, we conducted an experiment using skin cells to confirm whether our materials were effective in enhancing ECM components.

After treating Hs68 with andrographolide and BPE or retinol, we measured changes in gene expression levels. We found that treatment with andrographolide and BPE significantly increased *ELN*, *FN1*, and *FBN1* gene expression compared to the non-treated group (Figure 3a). However, there was no significant difference between the combined treatment group and retinol group except for *FN1* gene expression (Figure 3a). Next, we tested fibrillin-1 protein levels by immunostaining. UV-B irradiation significantly reduced fibrillin-1 protein levels in Hs68 (Figure 3b,c). However, treatment with andrographolide and BPE effectively restored fibrillin-1 protein levels, which were reduced by UV exposure (Figure 3b,c). These results comprehensively suggest that andrographolide and BPE enhances ECM, contributing to the overall improvement in skin.

### 3.4. Anti-Oxidant and Anti-Inflammatory Effects of Combined Treatment of Andrographolide and BPE

The anti-oxidant system plays a vital role in protecting cells from damage caused by free radicals, such as ROS. A decline in anti-oxidant defense is often associated with aging [48]. Oxidative stress—particularly from UV exposure—contributes significantly to skin aging [49]. Recently, RXR agonists haven been shown to prevent cell death by reducing cellular ROS levels [50]. Based on this, we assessed whether andrographolide and BPE could stimulate the anti-oxidant system.

As oxidative stress, such as that induced by UV or H_2_O_2_, is known to cause cell death, we first conducted a cellular viability assay [51]. UV or H_2_O_2_ treatment induced cell death (Figure 4a,b). However, co-treatment with andrographolide and BPE successfully prevented UV- or H_2_O_2_-induced cell death (Figure 4a,b). To further understand the protective efficacy of andrographolide and BPE, we measured the cellular ROS levels. H_2_O_2_ treatment elevated ROS levels in these cells (Figure 4c); however, co-treatment with andrographolide and BPE significantly lowered cellular ROS levels induced by the H_2_O_2_ treatment (Figure 4c). These results indicate that the co-treatment of andrographolide and BPE has potential as a therapeutic agent for mitigating the detrimental effects of oxidative stress on cellular viability. These results are consistent with a previous study showing that RXR agonists can effectively reduce cellular ROS levels and prevent cell death [50].

Skin inflammation is an immune response triggered by various internal and external causes, causing skin irritations, such as redness, pain, itching, and dryness [52,53]. Keratinocytes in the skin play a central role in the immune response by producing several interleukins, including interleukin-1 (IL-1) and 6 (IL-6) [54]. Excessive interleukin production, whether from internal or external stimuli, can lead to skin inflammation [55]. Therefore, we conducted experiments using keratinocytes to evaluate the anti-inflammatory effects of andrographolide and BPE.

IL-6, produced by skin keratinocytes, plays an important role in cutaneous immune response [56]. Previous studies have linked the overexpression of IL-6 with psoriatic skin [57]. Therefore, we first analyzed *IL-6* gene expression levels using HaCaT after UV irradiation, which is known to induce aging and increase skin inflammation [58]. We found that UV irradiation increased *IL-6* gene expression; however, treatment with andrographolide and BPE significantly recovered the UV-induced *IL-6* gene expression (Figure 4d). Interleukin-1 alpha (IL-1α), another cytokine secreted by keratinocytes, is involved in an acute immune response in the skin [56]. As IL-1α overexpression is associated with skin disease and irritation, we also measured IL-1α levels secreted by HaCaT through ELISA. While retinol treatment did not prevent UV-induced IL-1α elevation, the co-treatment of andrographolide and BPE successfully decreased UV-induced IL-1α secretion (Figure 4e).

In these experiments, we demonstrated the effectiveness of andrographolide and BPE in reducing oxidative stress and UV-induced inflammation. However, the specific mechanisms and signaling pathways responsible for these beneficial effects were not investigated in this study. Further research is necessary to gain a deeper understanding of the underlying mechanisms and to develop more potent and efficient cosmetic treatments.

### 3.5. Necessity of RXR Activation for Efficacies of Andrographolide and BPE

Our results revealed that the combined treatment of andrographolide and BPE successfully activated RXR alpha and showed significant skin-related improvements in both Hs68 and HaCaT. Although these findings suggest a correlation between RXR alpha activation and the efficacy of andrographolide and BPE, they did not prove that RXR alpha activation is responsible for the complex’s effects. To address this, we investigated whether RXR activation is essential for the complex’s efficacy using RAR and RXR inhibitors.

We first performed the ELISA to evaluate collagen secretion. As expected, the co-treatment of andrographolide and BPE enhanced pro-collagen type I secretion (Figure 5a). However, when cells were co-treated with both RXR inhibitors, the ability to promote collagen enhancement was hindered (Figure 5a). Moreover, the RAR inhibitors did not impact collagen secretion (Figure 5a). Taken together, these results indicate that our materials require RXR activation to enhance collagen secretion.

Next, we investigated whether the anti-oxidant activity of andrographolide and BPE depends on RXR activation. Interestingly, we observed that the presence of an RAR inhibitor had no effect on anti-oxidant efficacy (Figure 5b). However, when cells were treated with an RXR inhibitor, cellular ROS levels significantly increased compared to andrographolide and BPE group (Figure 5b). Additionally, cellular viability was reduced to levels similar to those observed in the UV-exposed control group (Figure 5c). These results underscore the importance of RXR activation in the anti-oxidant activity of andrographolide and BPE.

To verify the effectiveness of these inhibitors, experiments were conducted to confirm their ability to inhibit receptor activation. We demonstrated that the RXR inhibitor effectively blocked RXR alpha activation induced by 9-cis retinoic acid, and the RAR inhibitor successfully inhibited retinol-induced RAR gamma activation (Appendix A). Furthermore, we treated the cells solely with inhibitors to evaluate any potential side effects. We observed that these inhibitors had no significant effect on collagen synthesis or cell viability (Appendix A), confirming that they functioned as expected without causing any significant adverse effects.

In conclusion, our study revealed that the efficacies of andrographolide and BPE in improving skin-related parameters, such as collagen secretion and anti-oxidant activity, are dependent on RXR alpha activation. These findings showcase the importance of andrographolide and BPE as a potential therapeutic agent for skin-related conditions. Further investigations are warranted to elucidate the downstream mechanisms by which RXR activation mediates the effects of andrographolide and BPE on skin improvement.

### 3.6. Wrinkle Improvement with a Cream Containing Andrographolide and BPE

We conducted a clinical trial in humans to investigate the efficacy of andrographolide and BPE in reducing wrinkles. Fourteen female volunteers, with a mean age of 34 years, were included in this study. All participants were asked to apply a cream containing andrographolide and BPE (RXR formula) on their faces for 4 weeks. Subsequently, changes in three types of wrinkles—fine wrinkles around the crow’s feet, under-eye wrinkles, and nasolabial folds—were measured using Antera 3D. Fine wrinkles around the crow’s feet and under the eye were evaluated using the Ra value, which is widely used as an indicator of fine wrinkle [59,60]. Nasolabial folds, which are deep skin folds extending from the sides of the nose to the corners of the mouth, were measured using the folded length parameter, a commonly used parameter for assessing nasolabial folds [61].

The topical application of the RXR formula led to notable improvements in fine wrinkles around the crow’s feet and under the eyes. Specifically, crow’s feet wrinkles improved by 8.89% and fine wrinkles under the eye by 5.62% (Figure 6a,b). Additionally, when comparing the baseline measurements taken at week 0, both regions showed statistically significant improvements in fine wrinkles after 4 weeks of cream application (Figure 6a,b). Moreover, the RXR formula markedly reduced nasolabial folds, with a 10.49% improvement when comparing the measurements taken at weeks 0 and 4 (Figure 6c).

### 3.7. Improvement in Elasticity and Hydration by RXR Formula Without Skin Irritation

To further investigate the effect of the RXR formula on the overall skin condition, we measured skin elasticity and hydration using Cutometer and Corneometer, respectively. We used the R2 and R7 parameters, which are widely recognized for assessing gross elasticity and skin firmness, respectively [62]. After 4 weeks of treatment, skin firmness improved by 43.51%, while skin elasticity increased by 21.89% compared to that at week 0 (Figure 7a,b). Furthermore, the RXR formula significantly enhanced skin hydration, with a higher rate of improvement than that in the retinol-treated group (Figure 7c). Overall, our findings suggest that the RXR formula effectively enhances various skin indicators.

Finally, we examined potential side effects of the RXR formula. Participants were asked to apply retinol cream or the RXR formula for 3 d and self-evaluate irritation scores across five categories. The RXR formula consistently caused less irritation than the retinol cream in all categories (Figure 7d). Additionally, when comparing overall irritation scores, the RXR formula demonstrated significantly lower levels of irritation than the retinol cream (Figure 7e).

In our clinical studies, we demonstrated that the RXR formula is efficient in improving various skin parameters, such as wrinkles, elasticity, and skin hydration. Notably, unlike retinol cream, which often causes skin irritation, the RXR formula did not show any signs of skin irritation. However, further research is needed to validate these promising results and explore the full potential of the RXR formula in skincare, including large-scale studies with bigger sample sizes and long-term assessments.

## 4. Conclusions

In the present study, we successfully discovered a novel RXR alpha-activating combination comprising andrographolide and BPE and highlighted its potential in RXR alpha activation. We confirmed its ability to promote collagen synthesis and enhance skin ECM components. Additionally, we discovered that the combined treatment of andrographolide and BPE possesses anti-oxidant and anti-inflammation properties that are closely associated with its ability to mitigate skin irritation. Moreover, the skin-enhancing effects of andrographolide and BPE were shown to depend on RXR activation. Based on these promising results from the in vitro experiments, we conducted human clinical trials. Our findings indicated that the RXR formula efficiently improved various types of wrinkles, enhanced skin elasticity, and boosted skin hydration while eliciting less irritation than retinol.

Despite these advantages, this study has certain limitations. We did not investigate the precise mechanisms and downstream pathways involved in RXR alpha activation to improve skin conditions. Additionally, RXR interacts with several heterodimeric molecules, such as the RAR, thyroid hormone receptor, and vitamin D receptor; however, we did not examine which of these partners are involved when RXR alpha is activated by andrographolide and BPE [63]. Further research is required to elucidate these mechanisms and advance our understanding of skin science for the development of efficient cosmetics.

Overall, the results of this study underscore the potential of the novel RXR alpha-activating materials as an effective and safe solution for improving skin health and appearance. These findings provide a more comprehensive understanding of skin science and pave the way for developing more efficient cosmetics in the future.

## Figures and Tables

**Figure 1 biomolecules-15-00296-f001:**
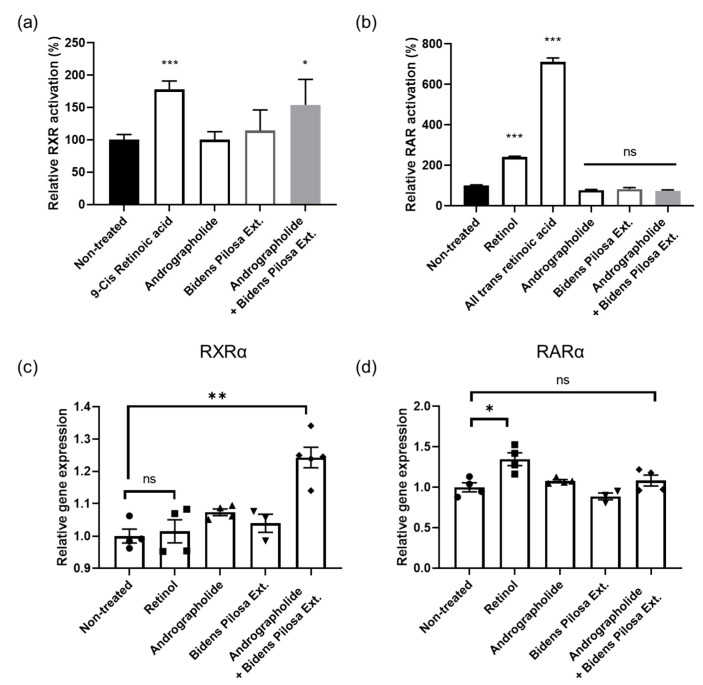
Effects of andrographolide and *Bidens pilosa* extract (BPE) on receptor activity and gene expression. (**a**) RXR alpha was activated by co-treatment of andrographolide and BPE. 9-cis retinoic acid was used as positive control (*n* = 6 per group). (**b**) RAR gamma was activated by retinol and all-trans retinoic acid. Andrographolide and BPE did not affect RAR gamma activity (*n* = 6 per group). (**c**) Expression of *RXRA* gene was enhanced by andrographolide and BPE (*n* = 3–6). (**d**) Expression of *RARA* gene was not affected by andrographolide and BPE (*n* = 3–4). Error bars represent standard error of the mean. ns: not significant, * *p* < 0.05, ** *p* < 0.01, *** *p* < 0.001; one-way ANOVA analysis and Kruskal–Wallis test for nonparametric statistics.

**Figure 2 biomolecules-15-00296-f002:**
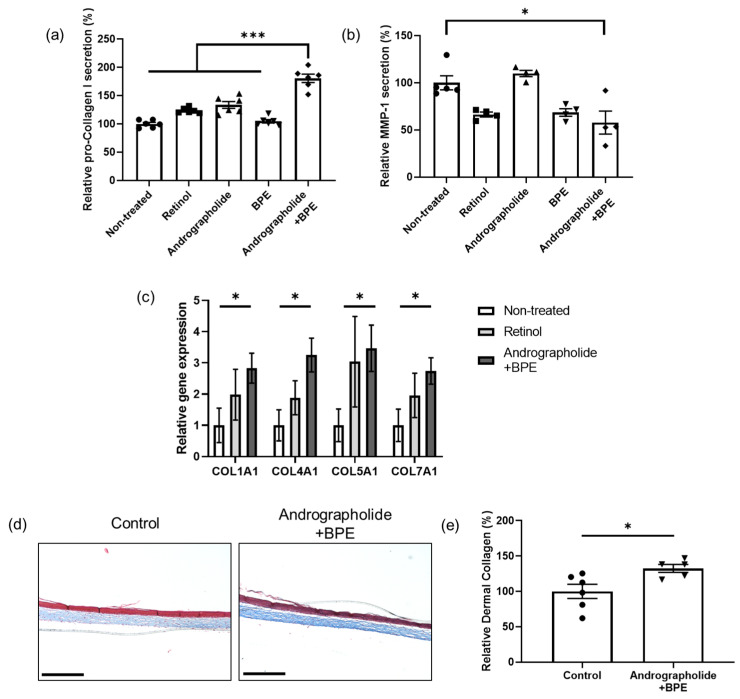
Effect of andrographolide and BPE on collagen synthesis in skin cells. (**a**) Protein level of pro-collagen type 1 alpha 1 secreted by skin cells (Hs68) was enhanced by combined treatment of andrographolide and BPE (*n* = 6 per group). (**b**) Protein level of MMP-1 secreted by skin cells (Hs68) was reduced by combined treatment of andrographolide and BPE (*n* = 4–5). (**c**) Gene expressions of different types of collagens were enhanced by co-treatment of andrographolide and BPE (*n* = 6 per group). (**d**) Representative images of artificial 3D skin. Scale bar = 275 μm. The blue color represents collagen. (**e**) Relative collagen content in dermal area of artificial 3D skin. Collagen content was enhanced by combined treatment of andrographolide and BPE (*n* = 5–6) Error bars represent standard error of the mean. * *p* < 0.05, *** *p* < 0.001; one-way ANOVA analysis, Student’s *t*-test and Kruskal–Wallis test for nonparametric statistics.

**Figure 3 biomolecules-15-00296-f003:**
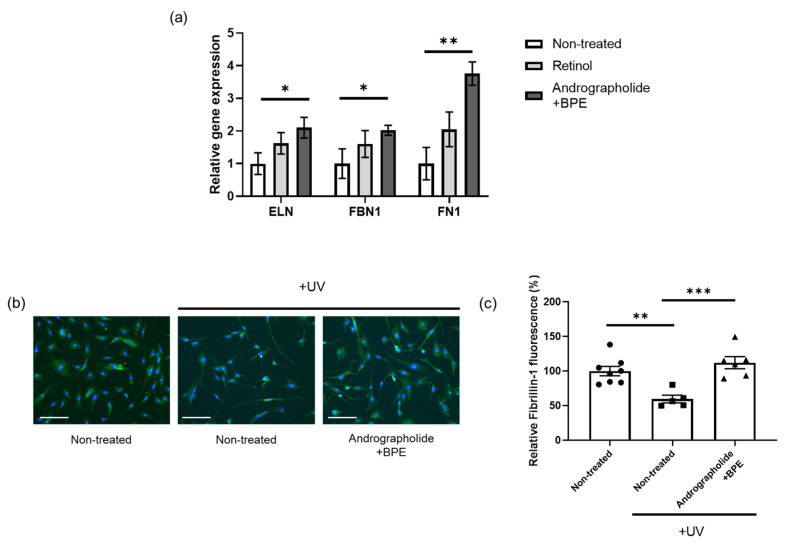
Effect of the combined treatment of andrographolide and BPE on ECM components. (**a**) Relative expression of ECM-related genes in Hs68 treated with active materials. *ELN*, *FBN1* and *FN1* gene expressions were enhanced by combined treatment of andrographolide and BPE (*n* = 6 per group). (**b**) Representative images of immunofluorescence staining for fibrillin-1 (Green). Nuclei were stained with DAPI. Scale bar = 125 μm. (**c**) Relative fluorescence intensity normalized to DAPI. Fibrillin-1 fluorescence intensity was enhanced by combined treatment of andrographolide and BPE (*n* = 5–8). Error bars indicate standard error of the mean. * *p* < 0.05, ** *p* < 0.01, *** *p* < 0.001; one-way ANOVA analysis and Student’s *t*-test.

**Figure 4 biomolecules-15-00296-f004:**
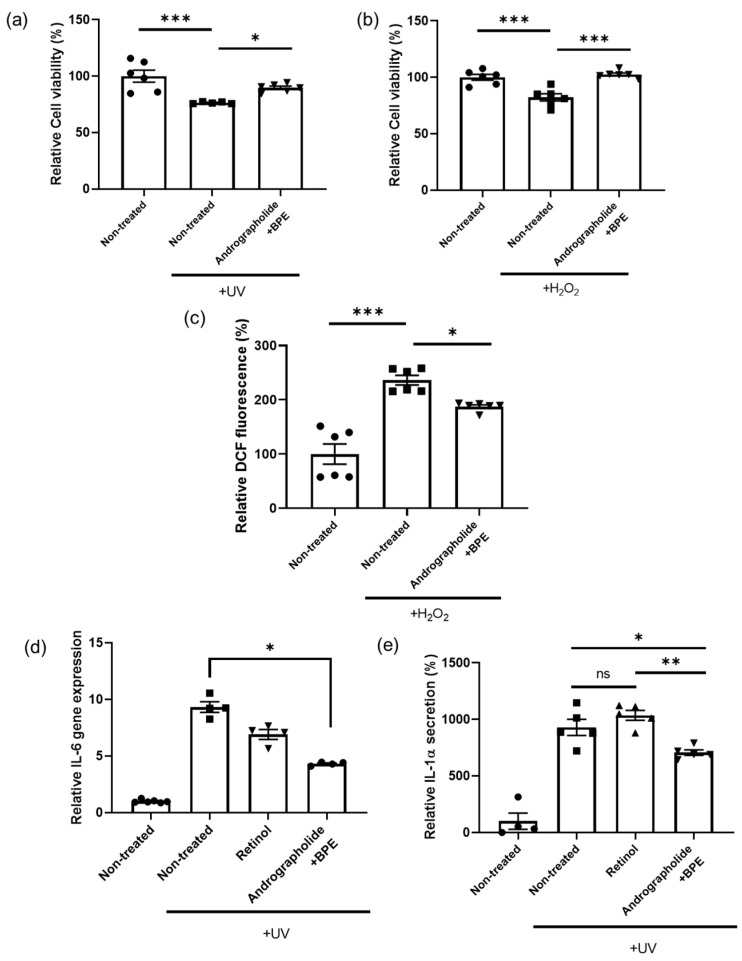
Analysis of anti-oxidant and anti-inflammation effects of andrographolide and BPE. (**a**) Effect of andrographolide and BPE on UV-induced cell toxicity. Combined treatment of andrographolide and BPE recovered UV-induced cell death (*n* = 5–6). (**b**) Effect of andrographolide and BPE on H_2_O_2_-induced cell toxicity. Combined treatment of andrographolide and BPE recovered H_2_O_2_-induced cell death (*n* = 6 per group). (**c**) Analysis of cellular reactive oxygen using DCFDA assay. Combined treatment of andrographolide and BPE reduced H_2_O_2_-induced cellular reactive oxygen (*n* = 6 per group). (**d**) Relative gene expression of *IL-6* in HaCaT was reduced by combined treatment of andrographolide and BPE (*n* = 4–6). (**e**) Relative IL-1α secreted by HaCaT was reduced by combined treatment of andrographolide and BPE (*n* = 4–5). ns: not significant, * *p* < 0.05, ** *p* < 0.01, *** *p* < 0.001; one-way ANOVA analysis and Mann–Whitney test for nonparametric statistics.

**Figure 5 biomolecules-15-00296-f005:**
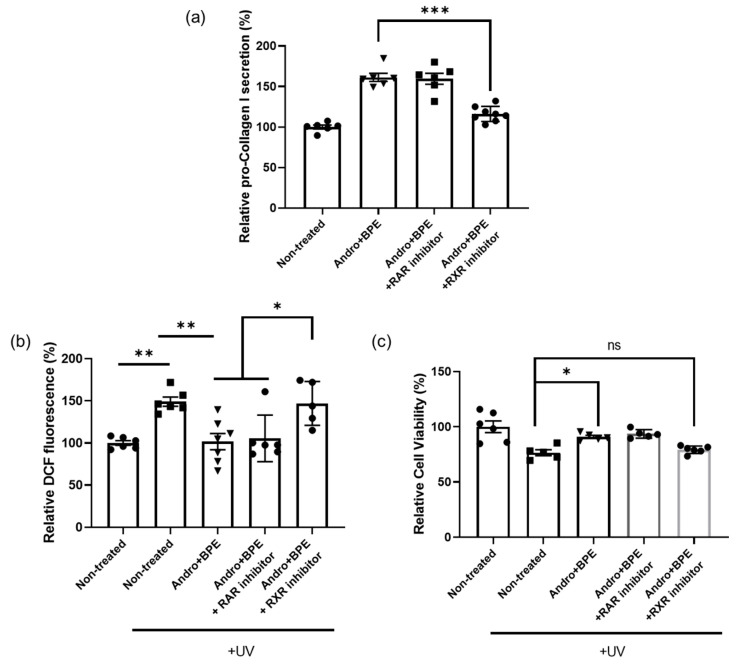
RXR activity-dependent collagen synthesis and anti-oxidant effects of andrographolide and BPE. (**a**) Effect of inhibitors on collagen synthesis efficacy of andrographolide and BPE. Collagen synthesis efficacy of andrographolide and BPE was hindered by RXR inhibitor (*n* = 6–8). “Andro” refers to andrographolide. (**b**) Anti-oxidant efficacy of andrographolide and BPE was reduced by RXR inhibitor (*n* = 5–7). (**c**) Increased cellular viability by andrographolide and BPE was reduced by RXR inhibitor (*n* = 5–6). ns: not significant, * *p* < 0.05, ** *p* < 0.01, *** *p* < 0.001; one-way ANOVA analysis.

**Figure 6 biomolecules-15-00296-f006:**
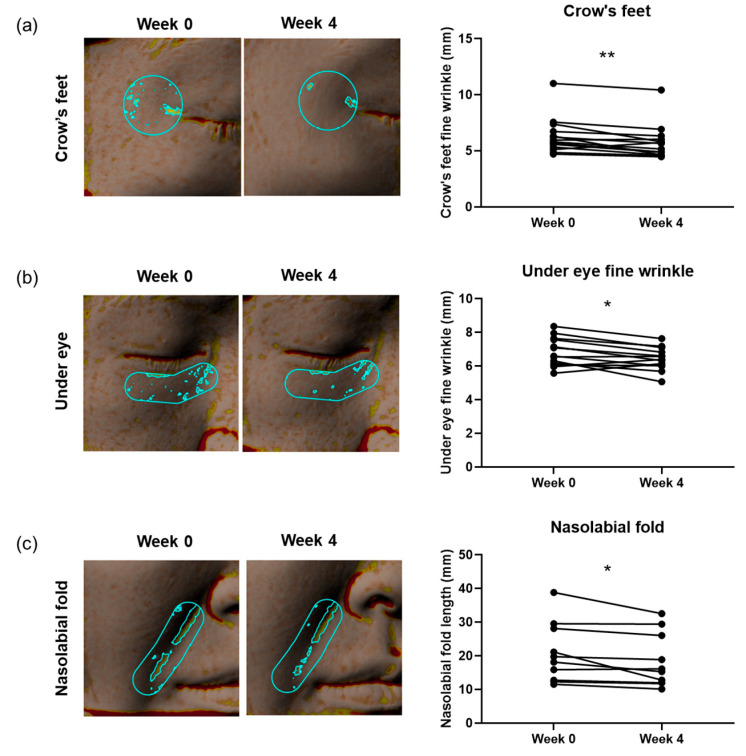
Wrinkle improvement by a cream containing andrographolide and BPE (RXR formula). (**a**) Fine wrinkle in crow’s feet was improved by RXR formula (*n* = 14). **Left**: representative images of crow’s feet captured using Antera 3D at weeks 0 and 4 after treatment. The highlighted area within the selected region represents fine wrinkles. **Right**: Change in crow’s feet fine wrinkle. (**b**) Fine wrinkle in under-eye area was improved by the RXR formula (*n* = 13). **Left**: representative images of under-eye area captured using Antera 3D at weeks 0 and 4 after treatment. The highlighted area within the selected region represents fine wrinkles. **Right**: Change in fine wrinkle of under eye. (**c**) Nasolabial fold length was reduced by the RXR formula (*n* = 10). **Left**: representative images of nasolabial fold captured using Antera 3D at weeks 0 and 4 after treatment. The highlighted area within the selected region represents nasolabial fold. **Right**: change in nasolabial fold length. * *p* < 0.05, ** *p* < 0.01; paired *t*-test.

**Figure 7 biomolecules-15-00296-f007:**
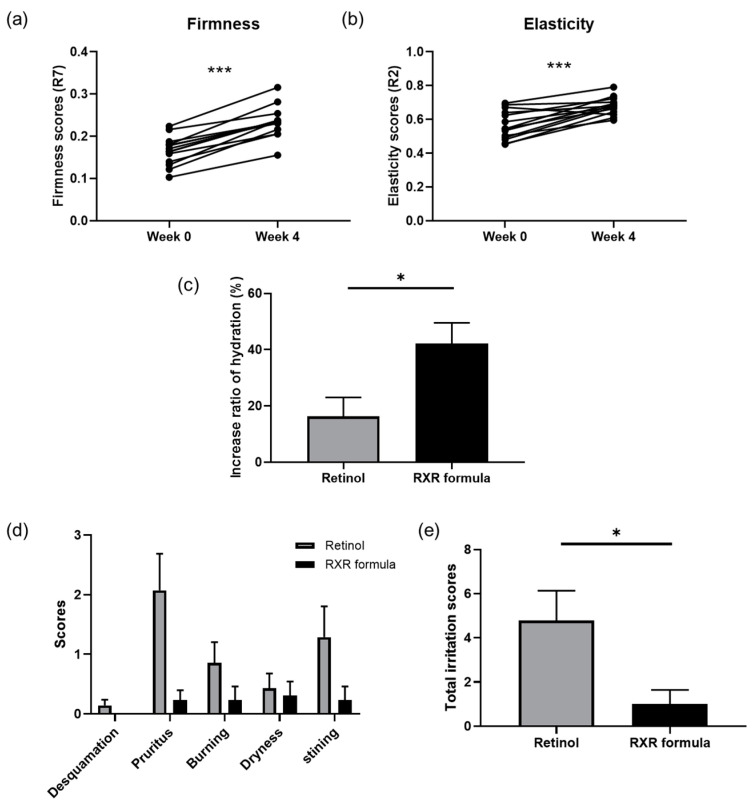
Measurement of skin elasticity, hydration, and irritation. (**a**) Skin firmness was improved 4 weeks after treatment of RXR formula (*n* = 14). Skin firmness was measured using R7 parameter. (**b**) Skin elasticity was improved 4 weeks after treatment of RXR formula (*n* = 14). Skin elasticity was measured using R2 parameter. (**c**) Comparison of skin hydration improvement rate between 0.1% retinol-treated and RXR formula-treated groups after 4 weeks (*n* = 7 for retinol group and *n* = 13 for RXR formula group). (**d**) Scores of each type of skin irritation for retinol- and RXR formula-treated groups (*n* = 14 for retinol group and *n* = 13 for RXR formula group). (**e**) Comparison of total irritation scores between 0.1% retinol-treated and the RXR formula-treated groups (*n* = 14 for retinol group and *n* = 13 for RXR formula group). * *p* < 0.05, *** *p* < 0.01; paired *t*-test, Student’s *t*-test and Mann–Whitney test.

## Data Availability

Data supporting the findings of this study are available from the corresponding author upon request.

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
