# Peer review of "Improvement of Skin Condition Through RXR Alpha-Activating Materials"

_biomolecules, 2025, doi:10.3390/biom15020296_

Round 1
Reviewer 1 Report
Comments and Suggestions for Authors
Dear Authors i went throug your manusccript wich represents a good scientific work. In my opinion the experimental work could be better elaborated in a better edition of the present manuscript.
The title has strengths, but it could benefit from a rephrase. “RXR activating complex” for instance is not completely supported by the evidence provided by this paper. It is to note that only RXR alpha seems hereby to be activated by the extracts used. Moreover, the synchronous administration of two compounds does not constitute a complex but a combined administration.
In addition to the above, the term RXR technology, often repeated in the manuscript, does not seem to be scientifically legit, since RXR technology may refer to various technologies around RXR such as chemical analysis technologies and thus, it is not specific. In my view RXR technology as a term cannot be accepted in this case.
In my view the abstract needs to be rewritten. Specifically:
Lines 9 – 24 According to journal’s instruction “The abstract should be a total of about 200 words maximum. The abstract should be a single paragraph and should follow the style of structured abstracts, but without headings” that means that the abstract should contain first the Background – and aims of the study. That means that conceptualization about RXR and RAR should be at the first part of the abstract before the aims of the study. In the Materials and methods, the complex must be better described. Subsequently the methods used should be listed (including the number of participants in the Clinical trial). Moreover, the summary of the results should be listed in the abstract. Then the conclusions should be listed clearly. The abstract, as it is now, is more review-like. A small concern also, in line 24 did you mean RAR rather than RXR?
The Introduction is very well written, and may need only minor amendments.
Line 41 Consider rephrasing. For example, instead of “photodegradation” it would be better to say “its potential photodegradation” to clarify that the degradation is not skin’s degradation but retinol’s degradation.
Lines 87 – 89 Are part of the conclusion of the study and they should be removed from the introduction.
Line 89 The aims of the study (null hypothesis) should be clearly described in a separate paragraph. (Why did you performed the study, what were you searching for or/and what was your null hypothesis)
A general observation is that in the materials and methods part you only search about RAR-gamma activation/suppression, thus, it would be better to be more precise in the Title/abstract/introduction part about RAR-gamma and not RAR in general that comprises also the alpha and beta RAR’s. The same observation also may be noted for RXR-alpha
In the Materials and Methods part, it is important to explain how many times the Cell lines experiments were executed and how many times each measurement was repeated.
Line 92 It would be better to report catalog numbers for the cell lines used to ensure reproducibility of the experiments and avoid confusion.
Lines 99 – 104 It is a bit confusing how you incorporated the water insoluble/poorly soluble compounds in the culture media. Please specify
Line 114 Specify the Luminescence reader and wavelength of measurement.
Lines 118 – 121 Please rephrase to clarify the procedure.
Line 128 Specify wavelength of measurement
Line 161 The test cream’s formulations shall be reported.
Line 199 It is very important to explain how the number of the participants was established. More precisely, in which statistical assumptions it was based.
Lines 200 – 201 The authors may want to list in their exclusion criteria the underaged persons and those who did not have the ability to read or write in Korean
Line 203 The formulations of the creams should be reported
Lines 223 – 227 In the data analysis part it is unclear how the investigators established the suitability of the tests implemented. Please report the Distribution normality test outcomes for each dataset an whether the tests performed were parametric or not. Please note that if the repetition of measurements in the cell lines was less than 5 the t- test may not suitable, but a non-parametric one (Mann Whitney U) may be more suitable. Please explain. Note also that for One side ANOVA you need also (exept from Shapiro Wilk) the equality of variances test (Levene’s) and the independence of measurements (not possible for multiple measurements of the same specimen)
Line 229 – 253 The authors should be more precise in the results they present. The RXR that they measured wasn’t the gamma one? Did they measure the RAR-gamma upregulation but the RAR alpha gene expression? Why did you do that? Please clarify and explain. Please do the same also, about the specific RXR upregulation and gene expression.
Lines 255 – 284 It is good to know what the extracts combination does to the fibroblasts but the claimed “Anti-Aging Effects of RXR Technology Through Collagen Enhancement” or Lines 285 – 310 “Through extracellular matrix (ECM) Component Enhancement” does not correspond to real worlds data. In contrast to the claimed activity the combined extract would never reach the fibroblasts (at least in doses that may exert the activity claimed). Considering also that the effectiveness claim is about cosmetic use, thus, for use to intact skin, the concern is much more important. In my view this subparagraph should undergo a careful rewrite, focusing on the real data achieved.
In Paragraphs 3.3, 3.4 and 3.5 The investigators used only two controls and the combined therapy, but they did not use the two extracts alone. Given that the two extracts possess antioxidant and anti-inflammatory properties of the one of the two extracts it is not clear whether the results achieved may be attributed to the extracts alone and not to the combined therapy. Please explain why you didn’t follow the same methodology as above
Note also that all the figures need to be Explained better. The captions should be in place to inform the reader about the content of the figure at a glance (without knowing the text). In some Diagrams also there are two labels for “control”. You need to specify which is positive and which negative. Finally, the number of repeated cultures/ measurements should be reported in each caption for each experiment.
Lines 326 – 329 belong to the discussion part. The Results part is only about results and the Discussion about what the results may show is a matter of the discussion part.
Lines 330 – 338 These lines fit better to the introduction.
Lines 348 – 352 belong to the discussion part.
Some observations about figure 6 and 7. It is unclear to how many patients these data refer. I was able to count only 8 – 9 dots in some diagrams. Moreover, observing the 7c it is unclear what this diagram presents in contrast to the subsequent 7d. If the 7 d shows that the skin dryness is not different between the two time points what do the hydration measurements show. The Corneometer, that was implemented, measures in reality skin conductivity that most of the times can be explained by skin hydration. But in this case that dryness is constant could also be attributed to oedema meaning that your “complex” may exert prooxidant action? Please explain.
The article lacks also a discussion part please add the discussion part after addressing the data analysis and all the other concerns.
After the addition / correction of all the above the Conclusions part will probably need to be rewritten thus this cannot be reviewed now.
I identified 4 out of 64 autocitations that in my view it is an acceptable rate.
Most of the literature cited is up to date an this constitutes a strength of the manuscript
Author Response
|
Response to Reviewer 1 Comments
|
||
|
1. Summary |
|
|
|
Thank you very much for taking the time to review this manuscript. Please find the detailed responses below and the corresponding revisions highlighted/in track changes in the re-submitted files.
|
||
|
2. Questions for General Evaluation |
Reviewer’s Evaluation |
Response and Revisions |
|
Does the introduction provide sufficient background and include all relevant references? |
Must be improved |
|
|
Is the research design appropriate? |
Can be improved |
|
|
Are the methods adequately described? |
Must be improved |
|
|
Are the results clearly presented? |
Must be improved |
|
|
Are the conclusions supported by the results? |
Not applicable |
|
|
3. Point-by-point response to Comments and Suggestions for Authors |
||
|
Comments 1: Dear Authors i went throug your manusccript wich represents a good scientific work. In my opinion the experimental work could be better elaborated in a better edition of the present manuscript.
|
||
|
Response 1: First of all, we deeply appreciate the reviewer’s efforts for giving insightful comments.
|
||
|
Comments 2: The title has strengths, but it could benefit from a rephrase. “RXR activating complex” for instance is not completely supported by the evidence provided by this paper. It is to note that only RXR alpha seems hereby to be activated by the extracts used. Moreover, the synchronous administration of two compounds does not constitute a complex but a combined administration.
|
||
|
Response 2: Thank you for pointing this out and we agree with this comment. Therefore, We decided to change the title as following: “Improvement of Skin Condition through RXR alpha-Activating Materials”
Comments 3: In addition to the above, the term RXR technology, often repeated in the manuscript, does not seem to be scientifically legit, since RXR technology may refer to various technologies around RXR such as chemical analysis technologies and thus, it is not specific. In my view RXR technology as a term cannot be accepted in this case.
Response 3: Thank you for pointing this out and we agree with reviewer’s critical comment. Based on reviewer’s comment, we decided to change the term ‘RXR technology’ to ‘andrographolide and BPE’, ‘co-treatment’ and ‘combined treatment’. We decided to label the cream containing andrographolide and BPE as ‘RXR formula’ in clinical trials and mark them in the abbreviation part.
Comments 3: In my view the abstract needs to be rewritten. Specifically: Lines 9 – 24 According to journal’s instruction “The abstract should be a total of about 200 words maximum. The abstract should be a single paragraph and should follow the style of structured abstracts, but without headings” that means that the abstract should contain first the Background – and aims of the study. That means that conceptualization about RXR and RAR should be at the first part of the abstract before the aims of the study. In the Materials and methods, the complex must be better described. Subsequently the methods used should be listed (including the number of participants in the Clinical trial). Moreover, the summary of the results should be listed in the abstract. Then the conclusions should be listed clearly. The abstract, as it is now, is more review-like. A small concern also, in line 24 did you mean RAR rather than RXR?
Response 3: Thank you for reviewer’s critical comments. Based on reviewer’s comments, we changed the Abstract part as following:
[page 1, line 9] ‘Retinol is well-known anti-aging material in the cosmetics industry owing to its proven superior efficacy both in vitro and in vivo. Despite its high efficacy, retinol is associated with limitations, such as skin irritation and its potential photodegradation. Retinol is converted into retinoid acid within cells, which then exerts a cellular response by activating both retinoic acid receptor (RAR) and retinoid x receptor (RXR). Noting that RAR activity is associated with skin irritation and RXR activation alone can enhance skin-related indicators without inducing inflammation, we developed an alternative approach for skin anti-aging focusing solely on RXR activation. We found that combined treatment of andrographolide and Bidens pilosa extract successfully activated RXR alpha and enhanced RXRA gene expression. Moreover, we investigated their efficacy using dermal fibroblasts and keratinocytes and found that they enhanced gene expression of extracellular matrix (ECM) proteins with anti-oxidant and anti-inflammation efficacies. Finally, in a human clinical trial, we confirmed that our materials successfully improved wrinkles in various areas, skin elasticity and hydration without causing irritating side effects. These findings highlight the potential of our RXR alpha-activating materials as an anti-wrinkle solution that avoids the typical side effects associated with retinol.’
Comments 4: The Introduction is very well written, and may need only minor amendments. Line 41 Consider rephrasing. For example, instead of “photodegradation” it would be better to say “its potential photodegradation” to clarify that the degradation is not skin’s degradation but retinol’s degradation. Response 4: Based on reviewer’s comments, we changed ‘photodegradation’ to ‘its potential photodegradation’.
Comments 5: Lines 87 – 89 Are part of the conclusion of the study and they should be removed from the introduction.
Response 5: Based on reviewer’s comments, we removed the sentence in lines 87-89.
Comments 6: Line 89 The aims of the study (null hypothesis) should be clearly described in a separate paragraph. (Why did you performed the study, what were you searching for or/and what was your null hypothesis)
Response 6: We appreciate the reviewer’s efforts for this critical comment. Based on reviewer’s comments, we elaborated the sentences about the aims of the study in Introduction part as following:
[page 2, line 84] “In this study, we explored whether combined treatment of andrographolide and BPE can activate RXR alpha and assessed the dependence of their efficacy on skin cell lines on RXR alpha activity. To investigate their mechanism of action and efficacies, we conducted experiments using skin cell lines, dermal fibroblasts, and keratinocytes and confirmed their skin-improving effects through a human clinical trial.”
Comments 7: A general observation is that in the materials and methods part you only search about RAR-gamma activation/suppression, thus, it would be better to be more precise in the Title/abstract/introduction part about RAR-gamma and not RAR in general that comprises also the alpha and beta RAR’s. The same observation also may be noted for RXR-alpha
Response 7: Thank you for pointing this out and we agree with this comment. In our experiments, we evaluated RAR-gamma and RXR-alpha activation. However, we vaguely referred to them as RAR and RXR without specifying which RAR or subtype of RXR they activate. Therefore, we clearly indicated the subtype in all parts of the manuscript according to the reviewer's comment.
Comments 8: In the Materials and Methods part, it is important to explain how many times the Cell lines experiments were executed and how many times each measurement was repeated.
Response 8: Based on reviewer’s critical comments, we inserted additional sentences in the Materials and Methods part as following:
[page 3, line 98] “Cell line experiments were executed at least three independent experiments.”
Comments 9: Line 92 It would be better to report catalog numbers for the cell lines used to ensure reproducibility of the experiments and avoid confusion.
Response 9: Based on reviewer’s comments, we added catalog numbers of the cell lines as following:
[page 3, line 91] “Human skin keratinocytes (HaCaT) and human skin fibroblasts (Hs68) were purchased from AddexBio (T0020001, San Diego, CA, USA) and the American Type Culture Collection (CRL-1635, Manassas, VA, USA), respectively.”
Comments 10: Lines 99 – 104 It is a bit confusing how you incorporated the water insoluble/poorly soluble compounds in the culture media. Please specify
Response 10: Based on reviewer’s critical comments, we added additional information as following:
[page 3, line 103] “All substances were diluted in DMSO to make a stock solution, then diluted in cell culture media and treated with cells.”
Comments 11: Line 114 Specify the Luminescence reader and wavelength of measurement.
Response 11: Based on reviewer’s critical comments, we inserted additional information about reader and measurement optics as following:
[page 3, line 118] “Luciferase activity was measured by luminescence fiber optics using a Synergy H1 microplate reader (BioTek Instruments Inc., Winooski, VT, USA).”
Comments 12: Lines 118 – 121 Please rephrase to clarify the procedure.
Response 12: Based on reviewer’s critical comments, we clarified the procedure in detail as following: [page 3, line 121] “Briefly, cells were seeded in black 96-well plates and incubated for 24 h. After overnight, the cells were treated with test compounds for 24 h. Then, LiveBLAzer™-FRET B/G (CCF4-AM) substrate included in the kit was added to each well. After 2 h of incubation at room temperature, fluorescence intensity was measured using a Synergy H1 microplate reader (BioTek Instruments Inc.). Fluorescence in the blue channel was measured with excitation filter (409/20 nm) and emission filter (460/40 nm). FRET signal in the green channel was measured with excitation filter (409/20 nm) and emission filter (530/30 nm). RXR alpha activity was quantified by dividing the blue emission values by the green emission values (460:530 nm).”
Comments 13: Line 128 Specify wavelength of measurement
Response 13: We added additional information about wavelength as following:
[page 4, line 135] “The concentration and purity of the extracted RNA were determined by the absorbance at wavelength 230, 260 and 280 nm using Nanodrop spectrometer (Thermo Fisher Scientific, Waltham, MA, USA).”
Comments 14: Line 161 The test cream’s formulations shall be reported.
Response 14: We added additional information about cream’s formulations as following:
[page 5, line 171] “The control cream formulation consists of following ingredients; Distilled water, tromethamine, carbomer (2-propenoic acid, polymer with 2,2-bis(hydroxymethyl)propane-1,3-diol 2-propenyl ether), xanthan gum, EDTA-3Na, 1,2-hexanediol, betaine, glycerin, dipropylene glycol, isocetyl myristate, dimethicone/vinyl dimethicone crosspolymer, cyclopentasiloxane, cyclohexasiloxane, squalene, caprylic/capric triglyceride, lecithin, C12-20 alkyl glucoside, C14-22 alcohols, beeswax, ceteareth-20, PEG-40 stearate, cetearyl alcohol, glyceryl stearate, stearyl alcohol and cetyl stearyl alcohol. Cream used in this study was formulated according to the protocols set by the LG H&H.”
Comments 15: Line 199 It is very important to explain how the number of the participants was established. More precisely, in which statistical assumptions it was based.
Response 15: Thank you for reviewer’s critical comments. To carry out our clinical trial, we enrolled the maximum number of participants we could find. We determined that 14 participants would be sufficient for our pilot study to assess the potential efficacy of our cream in a clinical study. Additionally, in the Results and Discussion section, we mentioned the need for a large scale study with an increased sample size in the future. In statistical analysis, we utilized a paired t-test to evaluate the effectiveness of the cream containing the RXR alpha-activating materials in improving wrinkles and elasticity, and an unpaired t-test to compare it with the retinol cream. We confirmed distribution normality test before t-test.
Comments 16: Lines 200 – 201 The authors may want to list in their exclusion criteria the underaged persons and those who did not have the ability to read or write in Korean
Response 16: Based on reviewer’s comments, we updated the exclusion criteria as following:
[page 5, line 218] “Pregnant women, those receiving skin treatments at clinics, minors and those who did not have the ability in Korean were excluded”
Comments 17: Line 203 The formulations of the creams should be reported
Response 17: Based on reviewer’s comments, we added additional sentence a as following:
[page 6, line 222] “The base cream was made as mentioned in section 2.5.”
Comments 18: Lines 223 – 227 In the data analysis part it is unclear how the investigators established the suitability of the tests implemented. Please report the Distribution normality test outcomes for each dataset an whether the tests performed were parametric or not. Please note that if the repetition of measurements in the cell lines was less than 5 the t- test may not suitable, but a non-parametric one (Mann Whitney U) may be more suitable. Please explain. Note also that for One side ANOVA you need also (exept from Shapiro Wilk) the equality of variances test (Levene’s) and the independence of measurements (not possible for multiple measurements of the same specimen)
Response 18: Thank you for reviewer’s critical comments. Based on reviewer’s considerate comments, we conducted statistical test to check if dataset follows a normal distribution. We conducted both Kolmogorov-Smirnov test and Shapiro-Wilk test because those tests are known to perform effectively even with small sample sizes and previous study showed that Shapiro-Wilk test is more powerful in normality analysis than other tests1). If the data followed a normal distribution, we used a t-test or one-way analysis of variance, otherwise we performed statistical analysis using a nonparametric test. We included the statistical analysis information in the figure legend. Moreover, we also included this information in the Material and Method statistical analysis part as following:
[page 6, line 245] “Before conducting the analysis for statistical significance, we first checked if the data followed a normal distribution. If the data followed a normal distribution, we used a t-test or one-way ANOVA analysis to determine statistical significance. If the data did not follow a normal distribution, we used a nonparametric test for our analysis.”
1) Razali, N. M., & Wah, Y. B. (2011). Power comparisons of shapiro-wilk, kolmogorov-smirnov, lilliefors and anderson-darling tests. Journal of statistical modeling and analytics, 2(1), 21-33.
Comments 19: Line 229 – 253 The authors should be more precise in the results they present. The RXR that they measured wasn’t the gamma one? Did they measure the RAR-gamma upregulation but the RAR alpha gene expression? Why did you do that? Please clarify and explain. Please do the same also, about the specific RXR upregulation and gene expression.
Response 19: Thank you for pointing this out. We confirmed through activation assay that our materials did not affect RAR gamma activity. Additionally, we wanted to check whether they affected not only RAR gamma but also other subunits, so we checked changes in RAR alpha subunit gene expression. In order to convey accurate experimental results, we included the following sentences:
[page 6, line 265] “To assess the impact of andrographolide and BPE on subunits of RAR other than RAR gamma, we verified any changes in the RARA gene expression and confirmed that these materials did not influence the expression of the RARA gene (Figure 1d).”
Comments 20: Lines 255 – 284 It is good to know what the extracts combination does to the fibroblasts but the claimed “Anti-Aging Effects of RXR Technology Through Collagen Enhancement” or Lines 285 – 310 “Through extracellular matrix (ECM) Component Enhancement” does not correspond to real worlds data. In contrast to the claimed activity the combined extract would never reach the fibroblasts (at least in doses that may exert the activity claimed). Considering also that the effectiveness claim is about cosmetic use, thus, for use to intact skin, the concern is much more important. In my view this subparagraph should undergo a careful rewrite, focusing on the real data achieved.
Response 20: We deeply understand reviewer’s concerns. Based on reviewer’s critical comment, we rephrased subtitle of 3.2 and 3.3 as following:
[page 7, line 286] “3.2. Collagen synthesis effects of Combined Treatment of Andrographolide and BPE in The Skin Cells” [page 8, line 325] “3.3. Efficacies of Combined Treatment of Andrographolide and BPE on Extracellular Matrix (ECM) Component Enhancement”
Comments 21: In Paragraphs 3.3, 3.4 and 3.5 The investigators used only two controls and the combined therapy, but they did not use the two extracts alone. Given that the two extracts possess antioxidant and anti-inflammatory properties of the one of the two extracts it is not clear whether the results achieved may be attributed to the extracts alone and not to the combined therapy. Please explain why you didn’t follow the same methodology as above
Response 21: Thank you for pointing this out. Our goal was to discover a non-irritating substance that activates RXR and has powerful efficacies. To achieve this, we confirmed that the combination of andrographolide and Bidens pilos extract activates RXR alpha and exhibits synergy in collagen synthesis. Based on these results, we selected the mixture of andrographolide and BPE as our final candidate, and conducted experiments to verify its efficacy. As the reviewer commented, the antioxidant or anti-inflammatory effects of this mixture could be attributed to either of the two substances, or may occur synergistically. However, what is important is that this mixture has antioxidant and anti-inflammatory effects.
Comments 22: Note also that all the figures need to be Explained better. The captions should be in place to inform the reader about the content of the figure at a glance (without knowing the text). In some Diagrams also there are two labels for “control”. You need to specify which is positive and which negative. Finally, the number of repeated cultures/ measurements should be reported in each caption for each experiment.
Response 22: Thank you for reviewer’s considerate comments. Based on reviewer’s comments, we modified the Figure and Figure Legend in more detail.
Comments 23: Lines 326 – 329 belong to the discussion part. The Results part is only about results and the Discussion about what the results may show is a matter of the discussion part. Lines 330 – 338 These lines fit better to the introduction. Lines 348 – 352 belong to the discussion part.
Response 23: We apologize for any misunderstanding. When writing our manuscript, we combined the Results and Discussion sections and mistakenly labeled it only as “Results.” We have now corrected the subtitle to “Results and Discussion.” This format is acceptable in Biomolecules, and papers in Biomolecules by other authors have also been written in the following format (Biomolecules 2025, 15(2), 204; https://doi.org/10.3390/biom15020204, Biomolecules 2025, 15(2), 181; https://doi.org/10.3390/biom15020181).
Comments 24: Some observations about figure 6 and 7. It is unclear to how many patients these data refer. I was able to count only 8 – 9 dots in some diagrams. Moreover, observing the 7c it is unclear what this diagram presents in contrast to the subsequent 7d. If the 7 d shows that the skin dryness is not different between the two time points what do the hydration measurements show. The Corneometer, that was implemented, measures in reality skin conductivity that most of the times can be explained by skin hydration. But in this case that dryness is constant could also be attributed to oedema meaning that your “complex” may exert prooxidant action? Please explain. Response 24: Thank you for reviewer’s critical comments. We conducted a clinical trial with a total of 14 subjects, and excluded participants who did not have initial wrinkle values in certain areas. For example, in the nasolabial fold area, no observations were made at week 0 for relatively younger subjects, and we excluded such cases from the analysis. We have provided detailed information regarding this matter in the Materials and Methods section as follows:
[page 6, line 241] “Participants who did not have any initial wrinkles in specific areas at week 0 have been excluded from the analysis.”
As mentioned in the Methods section, the ‘dryness’ shown in Figure 7d is one of the subjective irritation indicators that participants felt during the initial 3 days of use. We believe that comparing this value with the skin hydration values measured at week 0 and week 4 of use would not be appropriate. When the value of ‘dryness’ was found to be below 1, based on self-assessment during the initial 3 days, we determined that there was not a significant irritation about dryness. Additionally, no other side effects such as oedema or discomfort about skin moisture were observed or reported by the participants during the clinical trial.
Comments 24: The article lacks also a discussion part please add the discussion part after addressing the data analysis and all the other concerns.
Response 24: Sorry for the confusion. As mentioned in response 24, we combined the Results and Discussion sections.
Comments 25: After the addition / correction of all the above the Conclusions part will probably need to be rewritten thus this cannot be reviewed now.
Response 25: We would like to thank you again for taking the time to review our manuscript. We revised the manuscript based on the reviewer's comments. Please review once again.
Comments 26: I identified 4 out of 64 autocitations that in my view it is an acceptable rate. Most of the literature cited is up to date an this constitutes a strength of the manuscript.
Response 26: Thank you for reviewing our manuscript in detail.
|
||
Reviewer 2 Report
Comments and Suggestions for Authors
The manuscript entitled "Novel RXR-Activating Complex: A Promising Anti-Aging Solution without Side Effect" describes the effects of a combination of andrographolide with Bidens pilosa extract (BPE) on human skin cells in vitro regarding parameters related to skin aging, as well as, on human skin in vivo during a clinical trial. The study and the results presented are interesting and support the possibility of using this combination in cosmetic products.
However, there are some points in the manuscript requiring further clarification:
1) Since the authors show that the combination called "RXR Technology" acts through activation of RXR, why do they compare its effects with retinol in all Figs. starting from Fig. 2? Retinol activates both RXR and RAR, so all parameters studied could be affected by any of the two receptors or both. Using the more specific ligand 9-cis retinoic acid as a means of comparison would be more relevant.
2) In Fig. 2c, the authors should state if any statistically significant differences exist between the effects of retinol and those of RXR Technology on the expression of the four collagen genes.
3) Similarly, regarding Fig. 3a, the authors state in lines 298-299 that "RXR Technology group induced slightly higher expression levels than the retinol group", however they should clarify whether this slight difference is statistically significant.
Moreover, there are several points in the Materials and Methods section, that require the attention of the authors:
- the authors add 0.01 mM CaCl2 in the DMEM used for HaCaT culture (line 98), however normally DMEM already contains approx. 1.8 mM CaCl2, so what is the point of adding externally this small amount ?
- in Table 1, the authors list the compounds used in the study and their concentrations, however they should also explain how did they choose these specific concentrations
- the authors should state the supplier company for the black 96-well plates (line 110) and whether these plates were tissue culture-treated
- the primers listed in Table 2 are not the ones described in the text above (lines 136-139), with the exception of GAPDH; the authors should complete Table 2 with all primers used in the study
- in section 2.4, it is not clear whether the treatment with the test compounds was performed in serum-containing or serum-free medium; since for RT-PCR this was performed in serum-free medium (lines 124-125), one would expect that the same would happen for ELISA, however it should be stated clearly
- the conditions of UV-B irradiation are not so clear, as well: in what medium were the cells during irradiation?
Comments on the Quality of English Language
Although the language of the manuscript is comprehensible, there are some mistakes, that need to be corrected. A couple of examples follows:
In line 161, the verb should be used in plural, i.e., instead of "was" the authors should use "were"
In line 298, the expression "are crucial role" is not legitimate; the authors should use the verbs "play" or "have"
Author Response
|
Response to Reviewer 2 Comments
|
||
|
1. Summary |
|
|
|
Thank you very much for taking the time to review this manuscript. Please find the detailed responses below and the corresponding revisions highlighted/in track changes in the re-submitted files.
|
||
|
2. Questions for General Evaluation |
Reviewer’s Evaluation |
Response and Revisions |
|
Does the introduction provide sufficient background and include all relevant references? |
Yes |
|
|
Is the research design appropriate? |
Can be improved |
|
|
Are the methods adequately described? |
Must be improved |
|
|
Are the results clearly presented? |
Can be improved |
|
|
Are the conclusions supported by the results? |
Yes |
|
|
3. Point-by-point response to Comments and Suggestions for Authors |
||
|
Comments 1: The manuscript entitled "Novel RXR-Activating Complex: A Promising Anti-Aging Solution without Side Effect" describes the effects of a combination of andrographolide with Bidens pilosa extract (BPE) on human skin cells in vitro regarding parameters related to skin aging, as well as, on human skin in vivo during a clinical trial. The study and the results presented are interesting and support the possibility of using this combination in cosmetic products.
|
||
|
Response 1: First of all, we deeply appreciate the reviewer’s efforts for giving insightful comments.
|
||
|
Comments 2: However, there are some points in the manuscript requiring further clarification:
1) Since the authors show that the combination called "RXR Technology" acts through activation of RXR, why do they compare its effects with retinol in all Figs. starting from Fig. 2? Retinol activates both RXR and RAR, so all parameters studied could be affected by any of the two receptors or both. Using the more specific ligand 9-cis retinoic acid as a means of comparison would be more relevant.
|
||
|
Response 2: Thank you for reviewer’s critical comments. We aimed to demonstrate that our complex has a similar efficacy to retinol, which is known to be effective in the cosmetics industry, but with less irritation. As mentioned in the introduction and referenced paper, the irritation caused by retinoid is associated with RAR activity. To show that our complex does not cause irritation by only activating RXR, we chose retinol, which can also activate RAR, as the control group for comparison. In addition, 9-cis retinoic acid cannot be used as a cosmetic ingredient. Our goal is to develop a cosmetic ingredient that is both non-irritating and effective in skin improvement. Therefore, we chose retinol, which is a usable cosmetic ingredient, as the control group and compared the efficacy and irritation of our complex to retinol in a clinical study.
Comments 3: 2) In Fig. 2c, the authors should state if any statistically significant differences exist between the effects of retinol and those of RXR Technology on the expression of the four collagen genes.
Response 3: Thank you for reviewer’s critical comment. We found that there was no significant difference between RXR technology and retinol on the expression of the four collagen genes. To convey this result clearly, we inserted the sentence as following:
[page 8, line 303] “There was no significant difference between combined treatment group and retinol treatment group (Figure 2c).”
Comments 4: 3) Similarly, regarding Fig. 3a, the authors state in lines 298-299 that "RXR Technology group induced slightly higher expression levels than the retinol group", however they should clarify whether this slight difference is statistically significant. Response 4: Thank you for reviewer’s critical comment. We only found significant difference between RXR technology and retinol on the expression of FN1 gene. To clarify the result clearly, we changed the figure 3a and inserted the sentence as following:
[page 9, line 337] “However, there was no significant difference between the combined treatment group and retinol group except FN1 gene expression (Figure 3a).”
Comments 5: Moreover, there are several points in the Materials and Methods section, that require the attention of the authors: - the authors add 0.01 mM CaCl2 in the DMEM used for HaCaT culture (line 98), however normally DMEM already contains approx. 1.8 mM CaCl2, so what is the point of adding externally this small amount ?
Response 5: Thank you for pointing this out. There was confusion when we were writing the manuscript. For HaCaT culture, we used customized DMEM, which does not have Calcium and added 0.01mM CaCl2 to the customized DMEM. We updated the Materials and Methods part as following:
[page 3, line 95] “HaCaT were cultured in a customized calcium free DMEM (Solbio, Seoul, Republic of Korea) supplemented with 10% FBS, penicillin-streptomycin (Gibco), 1 mM sodium pyruvate (Gibco), 2 mM L-glutamine (Gibco), and 0.01 mM CaCl2 (Sigma-Aldrich, St Louis, MO, USA) at 37 ℃ with 5% CO2.”
Comments 6: - in Table 1, the authors list the compounds used in the study and their concentrations, however they should also explain how did they choose these specific concentrations
Response 6: Thank you for reviewer’s critical comments. When it comes to andrographolide and Bidens Pilosa extract, we performed several experiments to confirm ideal concentrations for enhancing skin-related indicators. We found that 1ug/ml concentration of andrographolide is most effective in enhancing collagen secretion in Hs68 cells. We screened the optimal combination ratio with Bidens Pilosa extract and found that 1:10 ratio was most effective combination ratio. Therefore, we set and performed all experiments with those concentrations. We also showed that the concentrations of the corresponding substances were working properly through RAR gamma and RXR alpha activation (Figure 1b, Figure S1 and Figure S2). Based on reviewer’s critical comment, we inserted the sentence in the Materials and Methods section as following:
[page 3, line 105] “The working concentrations of the materials used in this experiment were determined through separate experiments.”
Comments 7: - the authors should state the supplier company for the black 96-well plates (line 110) and whether these plates were tissue culture-treated
Response 7: Based on reviewer’s comments, we updated the information about black 96-well plates as following:
[page 3, line 114] “The cells were seeded in black 96-well plates (33396, SPL life sciences, Pocheon, Korea) and incubated overnight before being treated with test compounds for 24 h.”
Comments 8: - the primers listed in Table 2 are not the ones described in the text above (lines 136-139), with the exception of GAPDH; the authors should complete Table 2 with all primers used in the study
Response 8: Thank you for reviewer’s considerate comments. In qRT-PCR experiments, we used both commercial TaqMan primers and designed primers. In the paragraph, we provided the name of the genes and its corresponding catalog number for the commercial TaqMan primer. Furthermore, we included a table (Table 2) listing the gene names and sequences for the designed primers that we used in our experiments.
Comments 9: - in section 2.4, it is not clear whether the treatment with the test compounds was performed in serum-containing or serum-free medium; since for RT-PCR this was performed in serum-free medium (lines 124-125), one would expect that the same would happen for ELISA, however it should be stated clearly
Response 9: Sorry for the confusion. Based on reviewer’s comments, we added additional information about experimental procedure as following:
[page 4, line 154] “Then, the test compounds were added to serum free medium and incubated with the cells for another 24 h.”
Comments 10: - the conditions of UV-B irradiation are not so clear, as well: in what medium were the cells during irradiation?
Response 9: Thank you for reviewer’s critical comment. We added additional information about UV-B irradiation as following:
[page 5, line 186] “The culture media was changed with PBS and cells were irradiated once with UV-B at an intensity of 30 mJ/cm2 using BIO-SUN irradiation system (Vilber Lourmat).”
|
||
|
4. Response to Comments on the Quality of English Language |
||
|
Point 1: Although the language of the manuscript is comprehensible, there are some mistakes, that need to be corrected. A couple of examples follows: In line 161, the verb should be used in plural, i.e., instead of "was" the authors should use "were" Response 1: Based on reviewer’s comment, we corrected it to match the grammar.
[page 5, line 177] “Test cream formulations containing andrographolide and BPE were topically applied, with a control cream lacking these extracts.”
Point 2: line 298, the expression "are crucial role" is not legitimate; the authors should use the verbs "play" or "have" |
||
|
Response 2: Based on reviewer’s comment, we corrected it as following:
[page 5, line 327] “For example, elastic fibers, majorly composed of elastin and fibrillin-1, play crucial role for maintaining the integrity and skin elasticity of the skin” |
||
Round 2
Reviewer 1 Report
Comments and Suggestions for Authors
Dear authors
Thank you for adressing my concerns.
Reviewer 2 Report
Comments and Suggestions for Authors
The revised version of the manuscript is now suitable for publication in Biomolecules.